# Pre-migration socioeconomic status and post-migration health satisfaction among Syrian refugees in Germany: A cross-sectional analysis

**Jan Michael Bauer**[1], **Tilman Brand**[2], **Hajo Zeeb**[2,3]*

**1** Department of Management, Society and Communication, Copenhagen Business School, Copenhagen, Denmark, **2** Leibniz Institute for Prevention Research and Epidemiology—BIPS, Bremen, Germany, **3** Health Sciences Bremen, University of Bremen, Bremen, Germany

* zeeb@leibniz-bips.de

**Data Availability Statement:** Data are collected by the German Institute for Economic Research (https://www.diw.de/en). The authors are not

## Abstract

### Background

The large increase in numbers of refugees and asylum seekers in Germany and most of Europe has put the issue of migration itself, the integration of migrants, and also their health at the top of the political agenda. However, the dynamics of refugee health are not yet well understood. From a life-course perspective, migration experience is associated with various risks and changes, which might differ depending on the socioeconomic status (SES) of refugees in their home country. The aim of this paper was to analyze the relationship between pre-migration SES and self-reported health indicators after migration among Syrian refugees. Specifically, we wanted to find out how their SES affects the change in health satisfaction from pre- to post-migration.

### Methods and findings

We used data from the 2016 refugee survey, which was part of the German Socio-Economic Panel (GSOEP). Although cross-sectional by design, this survey collected information referring to the current situation as a refugee in Germany as well as to their situation before migration. Using a sample of 2,209 adult Syrian refugees who had entered Germany between 2013 and 2016, we conducted a cross-sectional and a quasi-longitudinal (retrospective) analysis. The mean ± SD age was 35 ± 11 years, with 64% of the participants being male. Our results showed a positive association between pre-migration self-reported SES and several subjective health indicators (e.g., health satisfaction, self-reported health, mental health) in the cross-sectional analysis. However, the quasi-longitudinal analysis revealed that the socioeconomic gradient in health satisfaction before migration was strongly attenuated after migration (SES-by-time interaction: −0.48, 95% CI −0.61 to −0.35, $p < 0.001$; unstandardized regression coefficients, 5-point SES scale and 11-point health outcome scale). Similar results were produced after controlling for sociodemographic characteristics, experiences during the migration passage, and the current situation in Germany. A sex-stratified analysis showed that while there was some improvement in health

legally permitted to share the data used for this study, but interested parties may contact the DIW representative Philipp Kaminsky to inquire about accessing this proprietary data (soepmail@diw.de). Further details about the data and access are provided here: https://www.diw.de/en/diw_01.c. 592731.en/soep.iab_bamf_soep_mig.2016. html#592737. The core file should allow and easy replication of our findings using the S1 Data Analyses and S2 Data Codebook files.

**Funding:** The authors received no specific funding for this work.

**Competing interests:** The authors have declared that no competing interests exist.

**Abbreviations:** BAMF-FZ, Research Centre of the Federal Office for Migration and Refugees; DIW, German Institute for Economic Research; FE, fixed effect; GSOEP, German Socio-Economic Panel; IAB, Institute for Employment Research; OLS, ordinary least squares; PHQ-4, 4-item Patient Health Questionnaire; SES, socioeconomic status.

satisfaction among men from the lowest SES over time, no improvement was found among women. A limitation of this study is that it considers only the first months or years after migration. Thus, we cannot preclude that the socioeconomic gradient regains importance in the longer run.

## Conclusions

Our findings suggest that the pre-migration socioeconomic gradient in health satisfaction is strongly attenuated in the first years after migration among Syrian refugees. Hence, a high SES before crisis and migration provides limited protection against the adverse health effects of migration passage.

---

## Author summary

### Why was this study done?

- Refugees are a vulnerable group because of the adversities and uncertainties they experience before, during, and after migration.

- It is not clear how the social rank in society before their flight affects the health of refugees when they arrive in the host country.

- Therefore, we analyzed the relationship between socioeconomic status before migration and (the change of) indicators of subjective health status among refugees.

### What did the researchers do and find?

- We found that a higher socioeconomic status was associated with a slightly better subjective health status after migration.

- However, we also found that the difference in health satisfaction between refugees of high and low socioeconomic status became much smaller after migration.

### What do these findings mean?

- Our findings provide new insight into the relationship between socioeconomic status and the health of refugees.

- A high socioeconomic status does not necessarily protect refugees from the negative influences during migration and the first months or years in the new country

## Introduction

According to the International Organisation for Migration, the estimated number of international migrants increased between 2000 and 2015 from 155 million to 244 million worldwide [1]. Since 2005, Germany hosts the second largest number of international migrants after the United States [2], and has recently witnessed a marked increase in the number of refugees and asylum seekers. While displaced persons such as refugees make up only a small proportion of all migrants [1], they deserve heightened attention because they are in a particularly vulnerable situation [3–5]. Among the migrants who recently arrived in Germany, refugees from Syria constitute the largest single group [6]. The large increase in numbers of refugees has put the issue of migration itself, the integration of migrants, and also their health at the top of the political agenda [7,8].

Migration can be viewed as a fundamental life-course transition that is often accompanied by a massive change in social and physical exposures. Therefore, Spallek and colleagues [9] suggest that exposures before, during, and after migration should be distinguished from each other. Before migration, exposures during critical or sensitive periods early in life, as well as driving factors of migration such as economic desperation, political oppression, war, or natural disasters play a role. During their journey, migrants, especially refugees, may be exposed to stark adversities when crossing deserts or the seas. They may become victims of trafficking, spend time in detention, or be exposed to physical or sexual violence. Shortly after arrival, access to better healthcare, sanitation, and nutrition may have a positive impact on migrants' health, but their legal status, housing conditions, economic prospects, and the contexts of reception may also play a role. With regards to healthcare, it has to be noted that access for refugees in Germany is restricted to some basic level, in many cases excluding treatment of chronic conditions until the individual refugee status has been clarified [10]. Thus, while this level of access to healthcare may mean improvement for some refugees, others who might have been able to afford high-standard care in their country of origin may find themselves temporarily—or in some cases for an extended period of time—in a worse situation.

The positive relationship between socioeconomic status (SES) and health has been found across a broad range of indicators and in diverse populations [11]. Even though the mechanisms behind the SES–health relationship are far from straightforward [12], one would expect that individuals with a higher SES before migration are better able to reduce or avoid risks during times of crisis and migration, which allows them to maintain a better health status for themselves and their families. These dynamics might have been amplified in the specific context of the crisis in Syria. Before the conflict, compared to other countries in the region, Syria could credit itself with having improved the living standards of its citizens over the last decades and providing them with an adequate healthcare system [13]. This progress was pushed back substantially by the civil war and subsequent economic sanctions. Job loss and rising prices for food and medicine resulted in serious health consequences, particularly for vulnerable groups and the chronically ill (i.e., children, pregnant women, and the elderly). Through various pathways, such negative health effects can be long-term [14] or even considered irreversible, particularly for the most vulnerable [15,16]. This prior evidence stipulates that a high precrisis SES of citizens might mitigate the health effects of a conflict, which could result in a widening of health disparities along the SES gradient even before the refugees have to leave their home countries.

However, status inconsistencies [17] or downward social mobility, which can occur due to migration, may affect refugees starting from a higher SES more strongly than those with a lower SES. For instance, the SES for those coming from very poor and deprived areas may improve or stay the same. On the other hand, others leave their country of origin where they had a high SES and may face great difficulties in trying to attain the same status in the host

country. There is evidence that some migrants with high educational degrees only find low occupational positions because their qualifications are not recognized in the host country [18]. Shortly after their arrival, refugees from different SES may find themselves in the same position regarding their housing situation in large refugee accommodations and their legal status, which does not allow them to seek employment. In other words, pre-migration SES differences are diminished and leveled to an overall low position.

Our analysis focuses on the relationship between pre-migration SES and post-migration health, specifically on the question of whether pre-migration SES moderates the change in health status from pre- to post-migration. To this end, we rely on large-scale survey data from recently arrived refugees from Syria in Germany, which provide a rich set of health indicators from different time periods, including the pre-migration period. Such retrospective data provide the opportunity to gain new insights into the complex health dynamics linked to the migration process, which is the main motivation for this study. Refugees are a vulnerable group and studying the underlying dynamics of their health trajectories can improve our ability to identify and help those particularly at risk. Additionally, these data capture the disruption of life courses through the Syrian crises and provide a unique opportunity to gain novel insights into the relationship between SES and health that are interesting from a scientific perspective and expand our current understanding. In summary, this study investigates two competing hypotheses: (1) A higher pre-migration SES allows refugees to maintain a better health throughout the migration period and within the destination countries. Hence, the migration process increases health disparities along the SES distribution. Alternatively, (2) pre-migration differences in health status across SES might be reduced after migration due to the leveling of SES and possibly status inconsistencies.

## Methods

This study is a retrospective, quasi-longitudinal analysis of a refugee survey that covers information about living circumstances and health before and after migration. The survey is part of the German Socio-Economic Panel (GSOEP) administered by the German Institute for Economic Research (DIW). The GSOEP is one of the longest running representative panel studies in the world that is frequently supplemented by other data with specific focus on areas within Germany. All data are publicly available for scientific research use. The present study relies on a subsample of the GSOEP reflecting the population of adult refugees predominately entering Germany between 2013 and the end of January 2016 [19]. As a response to the growing number of refugees in late 2015, the data were collected in collaboration with the Institute for Employment Research (IAB) and the Research Centre of the Federal Office for Migration and Refugees (BAMF-FZ). The dataset contains information about sociodemographics, health, attitudes, and living conditions in the migrants' country of origin. The sampling was based on the German Central Register of Foreigners (Ausländerzentralregister) using a multistage clustered sampling design. Data were collected through face-to-face interviews in seven different languages. Individuals selected from the register represented so-called "anchor respondents" for their household and answered the survey for themselves as an individual and as a representative for the household. Furthermore, all household members above 18 years were interviewed, while information about children and adolescents was captured through proxy data provided by the adult household members. The response proportion was 48.6% [19]. A detailed description of the survey methodology is available elsewhere [20]. The data can be accessed for scientific purposes upon registration on the DIW's website [21].

The total refugee sample consists of 4,527 individuals from more than 20 countries. In this analysis, we exclusively focus on participants with a Syrian nationality, who make up nearly

half of all refugees. The focusing on the Syrian sample helps reduce potential confounding by minimizing differences in the time line of the migration history and culture between refugee groups from different countries. We excluded 20 individuals who indicated that they entered Germany prior to 2013, leaving a core sample of 2,209 adults (18 years and older). Even though this dataset is cross sectional by design, the questionnaire collects information from different time periods. Questions referred to the current situation as a refugee in Germany (T1) or to the situation before the crisis in Syria (T0). For several constructs, the same questions were asked for both time periods.

## Health indicators

In our analysis we used five indicators for health and relied on health satisfaction as an indicator for subjective health status in our main outcome. Health satisfaction was measured with a single question for the current situation as well as retrospectively for the situation before "the crisis, the war or the conflict in [the] country of origin." For the latter, the participants were asked, "How satisfied were you with your health at that time?" with the possible responses ranging from 0 "totally dissatisfied" to 10 "totally satisfied." The question was mirrored with specific reference to the current situation, "How satisfied are you with your current health?" and had the same response scale. The survey also assessed general life satisfaction in a similar way, asking about life in general rather than health specifically. Additional health measures, which were only collected at T1, were the self-rated health state (five-point item from "poor" to "very well") and a measure of worries about health (three-point scale from "No, I don't worry at all" to "Yes, I worry a lot"). Furthermore, mental health was measured with the 4-item Patient Health Questionnaire (PHQ-4), which has been shown to reliably and validly measure depression and anxiety in the general population [22]. The longer version (PHQ-9) has also been validated in diverse migrant populations [23]. We inverted all scales where higher values corresponded to worse health outcomes.

We provide a correlational and graphical analysis of all constructs in the supporting information (S1 Table and S1 and S2 Figs). The construct validity of health satisfaction as our main dependent variable has been shown between different cohorts spanning over 60 years [24]. As S1 Table shows, health satisfaction was highly correlated with self-rated health (r = 0.82), which has been widely used as an indicator for health status in population-based studies and has proven to be a valid predictor of objective health measures such as mortality [24,25]. Health satisfaction was also correlated with the PHQ-4, showing that the measure of health satisfaction is related to symptoms of depression and anxiety.

## SES before migration

SES was assessed using a subjective measure of economic rank in the time before "the crisis, the war or the conflict." The question used was, "How would you estimate your financial situation at that time with the income of other people in your country?" and had five response options, ranging from "well below average" (0) to "well above average" (4). Such subjective measures of SES go back to Adler and colleagues [26], have been validated in cross-country studies [27], and have also been used in a number of studies linking subjective SES to health outcomes [11,28–30]. Measures of subjective SES have been shown to predict health outcomes independent of objective characteristics such as income and education [27,31,32]. Despite some evidence to the contrary [33], in a recent meta-analysis, an independent association between the most-used measures of subjective SES and physical health was observed [34]. The authors, however, also highlighted some methodological challenges. When SES and health outcomes are both self-reported, relationships might be inflated due to common method variance,

**Table 1. Characteristics of Syrian refugees included in this study.**

| Characteristics | Full sample | Sex-stratified sample | |
|---|---|---|---|
| | | Female | Male |
| *Individual measures* | | | |
| **Sex** | | | |
| Female | 810 (36%) | | - |
| Male | 1,399 (64%) | - | |
| **Age (years, mean)** | 34.5 (10.6, 18–75) | 34.8 (10.2, 18–72) | 34.4 (10.9, 18–75) |
| **Marital status** | | | |
| Single | 582 (26%) | 95 (12%) | 487 (35%) |
| Married | 1555 (71%) | 665 (82%) | 890 (64%) |
| Divorced | 35 (2%) | 20 (2%) | 15 (1%) |
| Widowed | 30 (1%) | 28 (3%) | 2 (<1%) |
| Number of children | 2.0 (2.2, 0–19) | 2.48 (2.1, 0–13) | 1.82 (2.2, 0–19) |
| *Precrisis measures/Syria (T0)* | | | |
| **Educational attainment at T0** | | | |
| Left with no qualifications | 526 (24%) | 195 (23%) | 331 (24%) |
| Middle school | 458 (20%) | 163 (21%) | 295 (22%) |
| Further practical-based | 195 (9%) | 56 (8%) | 139 (10%) |
| Further general-based | 635 (29%) | 242 (31%) | 393 (29%) |
| Other certificate | 61 (3%) | 21 (3%) | 40 (3%) |
| Educational details N/A | 334 (13%) | 133 (14%) | 201 (13%) |
| **Income in T0** | | | |
| 1st quartile | 313 (14%) | 93 (11%) | 220 (16%) |
| 2nd quartile | 310 (14%) | 69 (9%) | 241 (17%) |
| 3rd quartile | 320 (14%) | 69 (9%) | 251 (18%) |
| 4th quartile | 297 (13%) | 20 (2%) | 277 (20%) |
| Income N/A | 969 (42%) | 559 (69%) | 410 (28%) |
| Health satisfaction at T0 (mean) | 8.5 (2.2, 0–10) | 8.6 (2.1, 0–10) | 8.4 (2.3, 0–10) |
| Life satisfaction at T0 (mean) | 7.6 (2.6, 0–10) | 7.8 (2.5, 0–10) | 7.5 (2.6, 0–10) |
| Subjective SES at T0 (mean) | 2.2 (1.0, 0–4) | 2.2 (.9, 0–4) | 2.2 (1.0, 0–4) |
| *Postcrisis measures/Germany (T1)* | | | |
| Health satisfaction at T1 (mean) | 7.9 (2.4, 0–10) | 7.7 (2.5, 0–10) | 8.1 (2.4, 0–10) |
| Life satisfaction at T1 (mean) | 7.3 (2.3, 0–10) | 7.5 (2.0, 0–10) | 7.1 (2.4, 0–10) |
| Mental health* at T1 (mean) | 9.0 (2.7, 0–12) | 8.8 (2.6, 0–10) | 9.1 (2.7, 0–10) |
| Worries about health* at T1 (mean) | 1.5 (0.7, 0–2) | 1.5 (0.7, 0–2) | 1.6 (0.7, 0–2) |
| Self-rated health at T1 (mean) | 4.0 (1.1, 1–5) | 3.9 (1.1, 1–5) | 4.0 (1.1, 1–5) |
| Unemployed at T1 | 2,011 (91%) | 778 (96%) | 1,233 (88%) |
| *Migration experience* | | | |
| Number of negative experiences (mean) | 0.7 (1.2, 0–7) | 0.51 (1.0, 0–7) | 0.77 (1.2, 0–7) |
| **Duration of migration** | | | |
| less than 1 year | 1,156 (52%) | 422 (52%) | 734 (52%) |
| 1 year | 381 (17%) | 155 (19%) | 226 (16%) |
| 2 years | 268 (12%) | 87 (11%) | 181 (13%) |
| 3 years | 145 (7%) | 62 (8%) | 83 (6%) |
| 4 year or more | 121 (5%) | 33 (4%) | 88 (6%) |
| Duration N/A | 138 (6%) | 51 (6%) | 87 (6%) |
| Feeling welcome | 3.6 (0.8, 0–4) | 3.6 (0.8, 0–4) | 3.6 (0.8, 0–4) |
| **Year of arrival** | | | |

(*Continued*)

**Table 1.** (Continued)

| Characteristics | Full sample | Sex-stratified sample | |
|---|---|---|---|
| | | Female | Male |
| 2013 | 112 (5%) | 54 (7%) | 58 (4%) |
| 2014 | 532 (24%) | 199 (25%) | 333 (24%) |
| 2015 | 1,447 (66%) | 494 (61%) | 953 (68%) |
| 2016 | 118 (5%) | 63 (8%) | 55 (4%) |

Cell numbers are absolute frequencies (%) or means (standard deviation, range).

Mental health is the sum score of the four PHQ-4 items. Descriptive statistics are based on data for the specific cells (maximum $n = 2,209$). Variables with an asterisk

(*) were inverted so that higher numeric values correspond to better health in all measures.

Abbreviations: N/A, not available; PHQ-4, 4-item Patient Health Questionnaire; SES, socioeconomic status.

correlated reporting bias, and confounding [32,34,35]. However, while these methodological concerns are important, prior research supports the construct validity of subjective social status measures [36] and concludes that "subjective social status reflects the cognitive averaging of standard markers of socioeconomic situation and is free of psychological biases" [37].

## Covariates

Sociodemographic covariates included age, sex, marital status, educational attainment (i.e., certificate obtained from the last school visited in the home country), and number of children. Last income and region of origin in Syria were included as variables referring to the context before migration. Furthermore, variables related to the migration process and the post-migration period were assessed. A count variable of negative incidences or events during the migration passage covering deception, shipwreck, getting mugged, physical and sexual violence, extortion, and spending time in detention was created. The duration of the migration passage in years as well as the duration of residence (year of arrival) in Germany were included. With reference to the post-migration period, we used current employment status and context of reception. The latter was assessed with a single question asking whether the respondents had felt welcome when they arrived in Germany. The question had five response options ranging from "not at all" (0) to "fully agree" (4). A summary of all variables is provided in Table 1.

## Analysis

As has already been stated, the main interest of this study is to identify the effect of refugee SES in their home country, Syria (T0), on their current health status in Germany (T1). In a first step, we estimate the effect of subjective SES on health (measured by various indicators). Using multivariate regression allows for the adjustment of covariates, such as sociodemographic information, migration experience, and current situation. The following equation describes the model:

$$health_{i,1} = a + \beta_1 ses_{i,0} + X_i + \varepsilon_i \tag{1}$$

The health of individual $i$ in time 1 (i.e., 2016 in Germany) is measured by five different indicators: health satisfaction, self-rated health, mental health, health worries, and life satisfaction. $\beta_1$ is the coefficient of interest, subjective SES in precrisis Syria, while the vector $X_i$ captures a set of covariates that refer to the individual or both time periods. The intercept is indicated by $a$ and $\varepsilon_i$ captures the error term.

Our second analytical step exploits the fact that our questionnaire measures the main outcome variable (i.e., health satisfaction) for both time periods (T0 and T1). This data structure allows for the construction of a quasi-longitudinal dataset based on participants' self-assessed health satisfaction at different time points. This approach provides a suitable way to address the problem of starting from a higher baseline value, which might obscure any decline when simply comparing health satisfaction according to SES at T1. Our modeling is built on a two-period panel dataset and uses a difference-in-difference approach to control for differences in health satisfaction at T0. For our main results, we analyze the change in health satisfaction between T0 and T1 by using SES as a continuous "treatment" variable that moderates potential differences between both periods. A nonparametric specification of SES is also presented to validate the assumption of this linear relationship. The estimation approach is explained through the following regression equation:

$$health_{i,t} = a + \beta_1 time_t + \beta_2 ses_{i,0} + \beta_3 (time_t * ses_{i,0}) + X_i + \varepsilon_{i,t} \qquad (2)$$

We estimate the health satisfaction (health) of individual $i$ at time $t$. Time $t$ is a dummy variable indicating 0 for T0 and 1 for T1. Self-reported socioeconomic status at T0 is used as an indicator for SES. Hence, $\beta_3$ captures an interaction term between SES and the change in health between the two time periods, while controlling for baseline differences and the change in health experienced by all refugees. Similar to Eq 1, $X_i$ captures a set of covariates, $a$ is the intercept, and $\varepsilon_{i,t}$ is the error term, which we cluster on the individual level. Throughout the analysis, we used pairwise deletion of cases with missing data, with the exception that we created "missing" categories for categorical variables with high shares of missing data (e.g., income) and included them in the analysis in order to retain a sufficient sample size. Due to the pairwise deletion, the numbers of observations between the statistical models vary. Even though our main outcome measures of health are mostly ordered variables, we rely on ordinary least squares (OLS) as this method has been shown to yield minor differences only, compared to specific ordered response models, and also eases interpretation [38]. Results from an ordered-logit model provide comparable results and are provided in the supporting information (see specific results for reference). We also present the results of an individual fixed-effects model for robustness. This model aims to capture time-invariant unobserved individual heterogeneity in our quasi-longitudinal analysis. Additional analyses to assess the consistency of the findings include separate analyses for men and women as well as repeated analysis with an alternative outcome variable (life satisfaction).

No prospective analytical protocol was created. The main research question about the importance of social economic status for Syrian refugee health was formulated prior to data inspection and originated from the theories discussed in the introduction. The authors quickly settled on the initial difference-in-difference approach using retrospective survey information and later supplemented the core results by further robustness and subgroup analyses based on collegial feedback (e.g., generic cross-sectional analyses) and reviewer comments (e.g., refined analyses of ceiling effects). These data provide robustness checks to highlight the link between our main outcome and more objective health measures, as well as the problem of potential ceiling effects.

## Results

### Descriptive statistics

We present the descriptive statistics of our main estimation sample in Table 1. The sample consists of 64% men, and the average age of participants was 35 years. While the study covers a broad age range (18–75 years), only a small proportion of the sample (10%) is older than 50

years. Furthermore, (unaccompanied) minors are not included. The participating Syrian refugees are predominately married (>70%) and have two children on average. Looking at different health indicators, on average the respondents report their self-rated health as "good," even though health satisfaction between T0 and T1 declined by 0.5 points on an 11-point scale. A similar development can be observed for life satisfaction. The participants report mild levels of mental health issues. Upon arrival, which happened mostly in the years 2014 and 2015, around 90% of refugees felt "totally" or at least "mostly" welcomed. Only minor differences can be observed between women and men. On average, women are more likely to be married and report lower income. They are also less likely to report information on income at T0. Furthermore, women mostly score lower on health indicators but report higher levels of life satisfaction. A comparison of the descriptive statistics by SES is provided in S2 Table, showing that objective SES measures (i.e., education and income) generally follow the subjective self-reports.

## Cross-sectional analysis

Following our analytical strategy, Table 2 presents the results of our cross-sectional regressions, measuring the effect of subjective SES in Syria on various health indicators measured after the migration experience in Germany. In Table 2, we add variables from Panels A to D that could mitigate the relationship of interest in chronological order, starting with characteristics set before the refugees' migration experience, followed by assessments of the migration period itself and variables capturing their experience in Germany. In Panel A, where we only control for age and sex, all outcome variables point towards a positive relationship between higher SES and health. This relationship is statistically significant for health satisfaction, self-rated health, and (the absence of) health worries. For these three indicators, the findings remain robust after adding different sets of covariates, including sociodemographic information, migration experience, and variables measuring the experience in Germany (see Panels B–D). Adding objective measures of SES such as education and income reduces the point estimates for health satisfaction and self-rated health but does not fully explain the self-reported SES–health relationship.

Mental health and life satisfaction do not show a significant relationship with SES in the most parsimonious model (Panel A). However, after adding sociodemographic variables and indicators for migration experience, both coefficients increase and become marginally significant. This runs against the expectation that higher SES is generally associated with a less troublesome migration experience. When looking more closely into the data, we find that individuals with higher SES are more likely to report experiencing financial exploitation and fraud during migration, which has negative effects on mental health, in particular (lower values on the inverted mental health score), and thereby attenuates the positive relationship between mental health measures at T1 and SES at T0 (see S3 Table).

Overall, the results shown in Table 2 support the notion that high precrisis SES is associated with some positive health outcomes among Syrian refugees in Germany. Results using an ordered-logit model support these findings and generally provide more precise estimates, rendering the positive relationship between SES and mental health as well as SES and life satisfaction statistically significant (see S4 Table). However, as this first analytical step only assesses the residual relationship after migration, the next analysis puts focus on the actual change in health outcomes by accounting for retrospective (i.e., T0) measures.

**Table 2. Cross-sectional analysis of the association between self-reported SES before migration (T0) and self-assessed health indicators (OLS regression analysis).**

| SES measure, descriptors | (1) | (2) | (3) | (4) | (5) |
|---|---|---|---|---|---|
| | Health Satisfaction | Self-rated Health | Mental Health | Health Worries | Life Satisfaction |
| *Panel A: Covariates: sex and age* | | | | | |
| SES at T0 | 0.18*** | 0.07*** | 0.09 | 0.06*** | 0.05 |
| | [0.06–0.29] | [0.02–0.12] | [−0.04 to 0.23] | [0.03–0.09] | [−0.05 to 0.16] |
| N | 2,152 | 2,152 | 2,010 | 2,141 | 2,140 |
| adjusted $R^2$ | 0.08 | 0.09 | 0.00 | 0.09 | 0.01 |
| *Panel B: Covariates: sociodemographics* | | | | | |
| SES at T0 | 0.16*** | 0.06** | 0.12 | 0.06*** | 0.09 |
| | [0.04–0.29] | [0.01–0.12] | [−0.03 to 0.26] | [0.03–0.09] | [−0.02 to 0.21] |
| N | 2,082 | 2,082 | 1,945 | 2,071 | 2,070 |
| adjusted $R^2$ | 0.08 | 0.10 | 0.01 | 0.09 | 0.03 |
| *Panel C: Covariates: sociodemographics + migration experience* | | | | | |
| SES at T0 | 0.17*** | 0.06** | 0.14* | 0.06*** | 0.10* |
| | [0.05–0.29] | [0.01–0.12] | [−0.01 to 0.28] | [0.03–0.09] | [−0.02 to 0.21] |
| N | 2,082 | 2,082 | 1,945 | 2,071 | 2,070 |
| adjusted $R^2$ | 0.09 | 0.10 | 0.02 | 0.09 | 0.03 |
| *Panel D: Covariates: sociodemographics + migration experience + experience in Germany* | | | | | |
| SES at T0 | 0.16*** | 0.06** | 0.14* | 0.06*** | 0.09 |
| | [0.04–0.29] | [0.01–0.12] | [−0.01 to 0.28] | [0.03–0.10] | [−0.03 to 0.20] |
| N | 2,065 | 2,065 | 1,936 | 2,058 | 2,057 |
| adjusted $R^2$ | 0.10 | 0.11 | 0.04 | 0.09 | 0.06 |

Coefficients are unstandardized regression coefficients. Covariates included in Panel A: sex, age, and age². Added in Panel B: marital status, income at T0, educational certificate in T0, number of children, and Syrian birth region dummies. Panel C: negative migration experience and duration of migration. Panel D: employment status at T1, feeling of welcome, and year of arrival. $N$ = number of individuals. CIs (95%) based on heteroskedastic robust standard errors in brackets.

*$p < 0.1$,

**$p < 0.05$,

***$p < 0.01$.

Abbreviations: OLS, ordinary least squares; SES, socioeconomic status.

## Quasi-longitudinal analysis

The results of our analysis of health changes, where we focus exclusively on the measure of health satisfaction available in both periods, are presented in Table 3. The estimates of the most parsimonious model (column 1) indicate that higher SES is associated with better health satisfaction at T0. Our model predicts a 0.6-point increase in health satisfaction for each step increase in SES, which translates into an approximately 3-point difference on an 11-point scale between the lowest and highest SES at T0. Given that we modelled the relationship between SES and both periods as an interaction term, the T1 dummy can be interpreted as the change from T0 to T1 for individuals in the lowest SES category (ranging from 0 to 4). Hence, refugees living "well below the average" SES report better health in their current situation compared to their lives in Syria. The negative interaction term, however, reveals that the increase in health satisfaction is exclusively experienced by individuals at the lower end of the SES ladder. A non-significant change in health is already shown for those in the second lowest category, and our models predict that individuals with an "average" SES or above will be worse off compared to the pre-migration measurements.

This relationship is best understood when looking at Fig 1, in which we replicated the findings from the regression analysis shown in column 4 of Table 3, but modeled SES as a

**Table 3. Retrospective analysis of change in health satisfaction before (T0) and after migration (T1) by self-reported SES (OLS regression analysis).**

| Independent variables | (1) | (2) | (3) | (4) | (5) |
|---|---|---|---|---|---|
| | OLS | OLS | OLS | OLS | FE |
| SES at T0 | 0.64*** | 0.63*** | 0.63*** | 0.61*** | |
| | [0.53–0.75] | [0.51–0.74] | [0.52–0.74] | [0.50–0.73] | |
| T1 | 0.50*** | 0.53*** | 0.53*** | 0.52*** | 0.49*** |
| | [0.17–0.82] | [0.19–0.86] | [0.19–0.86] | [0.18–0.85] | [0.16–0.82] |
| SES × T1 | −0.48*** | −0.49*** | −0.49*** | −0.49*** | −0.47*** |
| | [−0.61 to −0.35] | [−0.62 to −0.36] | [−0.62 to −0.36] | [−0.62 to −0.35] | [−0.60 to −0.34] |
| Sociodemographics | No | Yes | Yes | Yes | No |
| Migration experience | No | No | Yes | Yes | No |
| Experience in Germany | No | No | No | Yes | No |
| *Number of observations* | 4,302 | 4,162 | 4,162 | 4,128 | 4,304 |
| adjusted $R^2$ | 0.09 | 0.10 | 0.10 | 0.11 | 0.07 |

Coefficients are unstandardized regression coefficients. Dependent variable for all regression is health satisfaction. Results in columns 1–4 are based on OLS. Covariates included in all regressions: sex, age, and age$^2$. Sociodemographics: marital status, income at T0, educational attainment at T0, number of children, and Syrian birth region dummies. Migration experience: negative migration experience and duration of migration. Experience in Germany: employment status at T1, feeling of welcome, and year of arrival. Column 5 based on within-estimator accounting for individual FE. CIs (95%) based on heteroskedastic robust standard errors clustered on the individuum in brackets.

*$p < 0.1$,

**$p < 0.05$,

***$p < 0.01$. A replication of this table with detailed covariates is provided in S5 Table and using an ordered-logit model is displayed in S6 Table.

Abbreviations: FE, fixed effect; OLS, ordinary least squares; SES, socioeconomic status.

categorical rather than as a continuous variable. This nonparametric modeling approach graphically confirms that treating SES as linear is an appropriate functional form. We can also see that at T1, health satisfaction is less steep along the gradient of SES, with only minor differences being observed between the SES extremes.

The other columns in Table 3 also show that our findings are robust when adding more control variables that could confound the observed pattern. Self-reported SES is correlated with other sociodemographic characteristics, such as objective SES measures (i.e., education and income) and family composition that we add in column 2. Additionally, we add a set of dummy variables to control for 14 different regions in Syria. With column 3, we enrich our model with information about the migration experience itself, which might also be correlated with SES and our health measure. As in column 2, the point estimates only change marginally. We add variables that describe the experience in Germany in column 4. This includes the current employment status and the experience of feeling welcome upon arrival. The latter shows a weakly significant positive relationship with health satisfaction. The overall SES–health relationship, however, remains unchanged.

Column 5 reports the results from a model using individual fixed effects (FE). Given that all our covariates are time invariant and we have no specific temporal reference point for T0 (i.e., we cannot calculate age differences), the FE model is completely unconditional. Accounting for unobserved heterogeneity does not significantly change our estimates and underlines the robustness of our main findings.

## Heterogeneity

The results presented in Table 3 are very robust between different model specifications and the addition of covariates. However, analyzing the data separately for men and women reveals

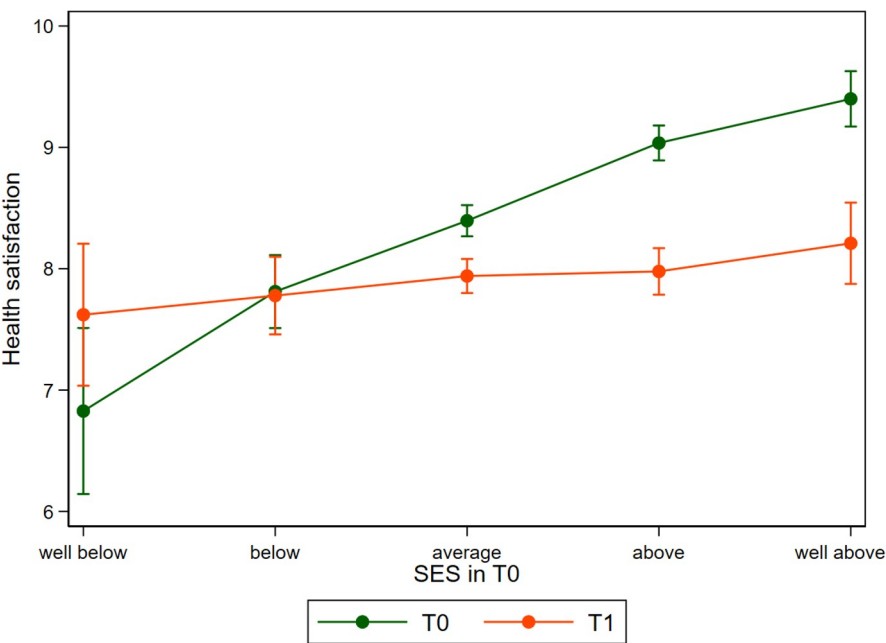

**Fig 1. Health satisfaction before (T0) and after migration (T1) in relation to self-reported SES before migration.**
Predictions derived from a linear model similar to column 4 of Table 3 but modelling SES as a categorical variable.
Error bars indicate 95% CIs based on heteroskedastic robust standard errors clustered on the individuum. SES,
socioeconomic status.

some noteworthy differences. Fig 2 is a graphical representation of the model displayed in column 4 of Table 3 for the male sample only, while Fig 3 reflects results for women (regression results provided in S7 Table). Even though the interaction term is similar in magnitude and statistically significant for both subgroups, the graphs differ remarkably. For men, we can see that satisfaction with health increases for refugees below the average SES, while those in the higher SES strata report lower health when compared to their time in Syria. For women, however, the levels of reported health satisfaction post-migration are not higher than at T0 for all SES strata. Women from the lowest SES strata show equal levels of health satisfaction, with increasing negative differences being observed for those with higher SES positions. For both men and women, the link between SES and health becomes substantially weaker compared to pre-migration measures.

## Life satisfaction

As the measure of life satisfaction is also available for both periods, we apply the same analytical approach used for health satisfaction for this more holistic well-being outcome measure (see S8 Table). The pattern of life satisfaction is similar but shows more pronounced effects compared to the health domain. The interaction effect for SES is strongly negative and life satisfaction decreases by 1 point on an 11-point scale (see Fig 4). These stronger results might be linked to the fact that general life satisfaction directly encapsulates more life domains such as family, career, or wealth that were severely affected by the conflict and lost through migration. Such factors might affect health only indirectly, resulting in smaller changes in our main outcome, which is measured on the same 11-point scale. Fig 4 displays these results in a nonparametric way and is based on the same regression as Fig 1, just using a different outcome variable.

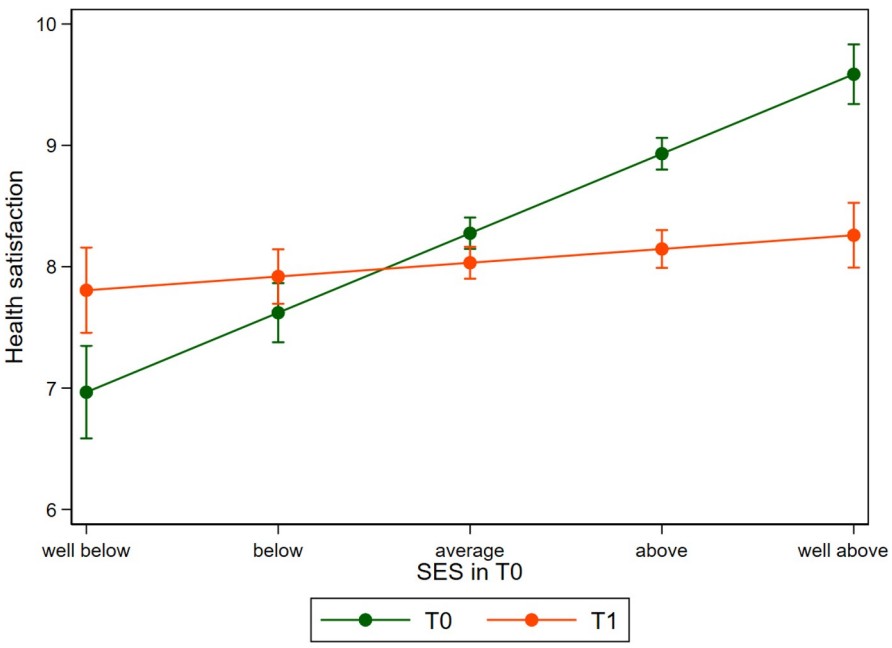

**Fig 2. Health satisfaction before (T0) and after migration (T1) in relation to self-reported SES before migration among men.** Predictions derived from a linear model for men are presented in S7 Table. Error bars indicate 95% CIs based on heteroskedastic robust standard errors clustered on the individuum. SES, socioeconomic status.

## Discussion

This study aimed to assess the effects of pre-migration and precrisis SES on health after migration in a sample of migrants from Syria who recently arrived in Germany. Adopting a life-course approach, the results of our study provide novel insights into the temporal relationship between SES and the health of refugees. Based on previous literature, we identified different pathways that could affect the health of refugees and therefore stipulated two competing hypotheses that might predict changes of health differences along the SES gradient.

Our main finding is that while pre-migration high SES in Syria has some remaining health benefits, differences in health satisfaction across SES while in Syria were reduced after migration. Health satisfaction improved among those with the lowest economic rank but became worse among those with an average or higher economic rank. This pattern remained robust after controlling for differences in migration experience and other potential confounders. Health satisfaction was highly correlated with self-rated health, which has proven to be a valid predictor of objective health measures. In addition, when using life satisfaction as an outcome variable, a similar pattern occurred. Furthermore, separated analyses for men and women revealed that health improvements were limited to those with the lowest health satisfaction before migration, i.e., men with the lowest SES. This is partly in line with mental health dynamics observed in migrant workers, where women showed a higher increase than men [39]. Sex heterogeneity has also been observed among refugees in Norway, where male refugees showed a higher likelihood to purchase antidepressants than non-refugee men, while differences among women were more pronounced for psychotropic medicine [40].

Other research has indicated that pre-migration SES may be positively related to post-migration health status. For example, a study among human trafficking returnees in Ethiopia reported that trafficked persons from families with a higher SES were less likely to display symptoms of posttraumatic stress disorders, indicating that financial resources of families were to some extent protective against the strict control of traffickers [41]. In contrast, our

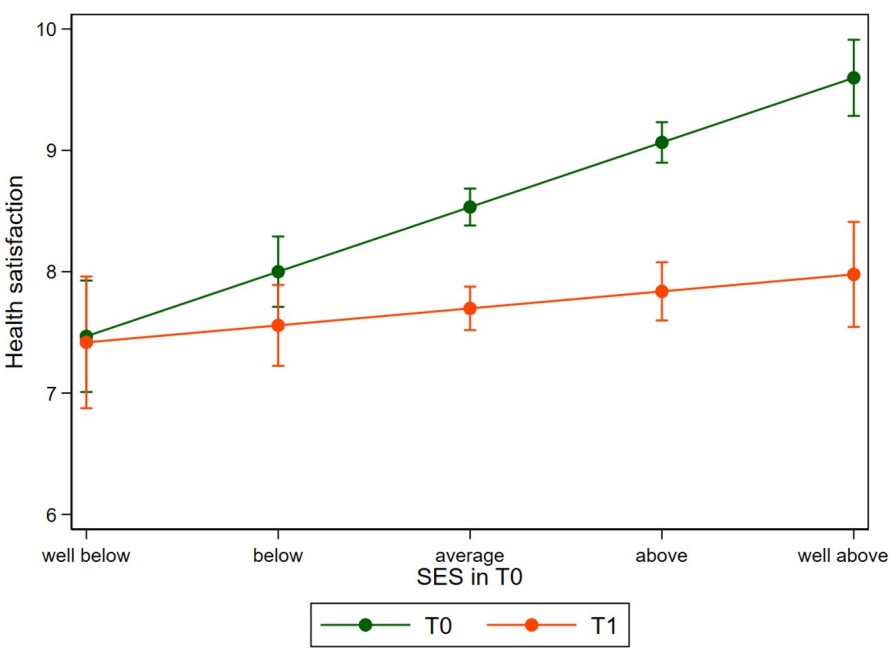

**Fig 3. Health satisfaction before (T0) and after migration (T1) in relation to self-reported SES before migration among women.** Predictions are derived from a linear model for women presented in S7 Table. Error bars indicate 95% CIs based on heteroskedastic robust standard errors clustered on the individuum. SES, socioeconomic status.

results show that refugees with a high SES from Syria are hardly able to maintain their better health status after arrival in Germany. This provides evidence against the fundamental cause hypothesis stating that individuals with a higher SES are always better able to protect their health [42]. Our data suggest that individuals with a higher SES experience different hardships during migration compared to people from lower SES and suffer more from extortion and fraud. This is, however, not the major driving factor for the observed dynamic as the time-by-SES interaction terms remain almost the same after controlling for migration experience. Another explanation could be that healthcare provision was unevenly distributed in precrisis Syria, offering high-quality (private) care in rich urban areas but only basic care in poor rural areas [43]. After arrival in Germany, Syrian refugees were exposed to the same restrictions in access to healthcare, regardless of their former SES. While this may have resulted in a loss in quality of care for those from higher SES, it could have meant an improvement in access to healthcare for those from the lowest SES. However, although access to healthcare may play a role, we do not believe that this can fully explain the leveling of the association between pre-migration SES and health satisfaction, because our study sample mainly consisted of middle-aged adults who may not require a lot of healthcare.

Apart from access to healthcare, one could argue that exposure to the same material living conditions in terms of housing and sanitation influences the leveling of the health gradient. Nevertheless, one could have expected that individuals from a formerly higher SES would be in a position to use their resources to improve their situation. Since this is not evident from our results, we would suggest that status inconsistency contributes to the explanation of our findings. Coming from a high SES in Syria may shape the expectations regarding living circumstances and position in society after migration. Finding themselves in refugee accommodations with limited access to the labor market, low levels of privacy, and unclear prospects about their residence permit may be experienced as a downward social mobility. According to the status

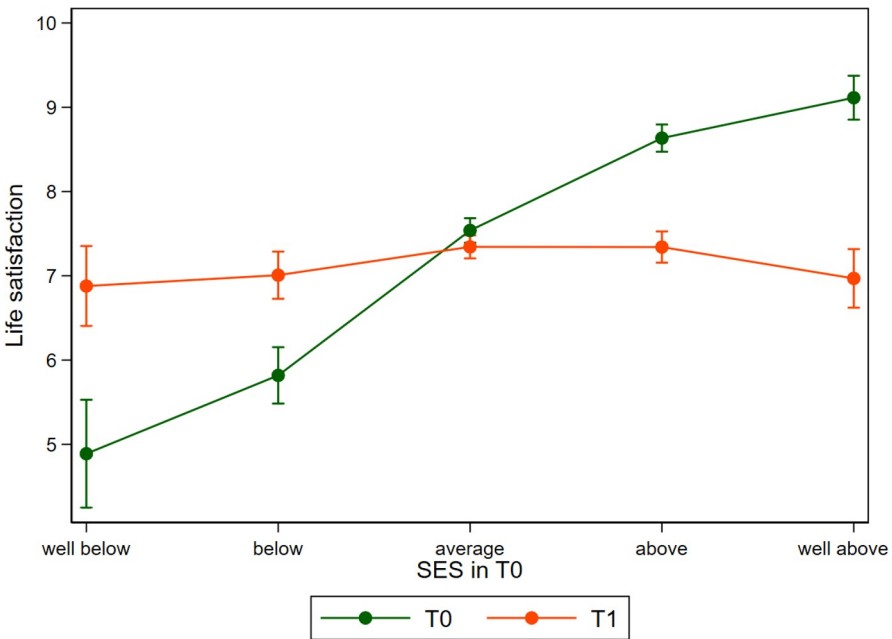

**Fig 4. Life satisfaction before (T0) and after migration (T1) in relation to self-reported SES before migration.**
Predictions derived from a linear model similar to column 4 of S8 Table but modelling SES as a categorical variable.
Error bars indicate 95% CIs based on heteroskedastic robust standard errors clustered on the individuum. SES,
socioeconomic status.

inconsistency theory, this could result in increased levels of mental health and physical symptoms, such as elevated blood pressure [44]. Although we cannot provide direct evidence within this study about which mechanism explains the reduction in health differential across premigration social status, we argue that both material and psychosocial factors play a role.

Looking into the different health indicators, we observe that refugees show little to no differences in the mental health indicator post-migration along the SES gradient, while SES retained some predictive power for more holistic indicators. This observation suggests that leveling within the psychological domain of health is particularly pronounced. Similar effects were found among Tongans migrating for work to New Zealand. Evidence from a natural experiment suggests that migration produces mental health benefits that are particularly pronounced in those with low mental health [25]. Although this type of migration substantially differs from the situation of Syrian refugees, these dynamics might alleviate SES differences. Rasmussen and colleagues [45] compared differences in mental health dynamics between refugees and a sample of voluntary migrants. While refugees initially come with a higher burden of psychiatric issues, the development of post-migration mental health issues was found to be similar in both groups over time.

The strong differences regarding life satisfaction with a change by one point of the 11-point scale for each point of the 5-point SES question emphasizes the importance of SES when studying well-being dynamics of migrants. The explicit consideration of SES might be helpful to explain the unclear results observed by prior studies such as the one conducted by Stillman and colleagues [46].

### Limitations

Despite the robust findings of this study, there are some limitations that need to be discussed. First, there is only a small proportion of older adults in our sample and (unaccompanied) minors—a group that is perceived as particularly vulnerable—were not included [47]. Therefore, the results of this study cannot be generalized to these groups.

Second, the quasi-longitudinal data are based on self-reported health measures at two time periods. As health satisfaction is measured on an 11-point scale, we have to consider potential ceiling effects. We can observe that 74% of individuals with the highest SES report the highest health satisfaction at T0. The share in the lowest SES level is only 34%. For these individuals, the measure does not allow the reporting of further increases in health satisfaction, which could introduce asymmetric bias of the change in the mean towards T1. This limitation is, however, unlikely to drive our results as we conduct several checks displayed in S9 Table: (1) The results remain robust for a subsample excluding all observations reporting the highest and lowest level of health satisfaction at T0. (2) After following a suggestion from the literature using a Tobit model to account for potential ceiling and floor effects [48], our main results remain robust for different specifications of the outcome variable.

Third, the retrospective assessment of precrisis health and SES does not refer to one specific point in time. Given a large set of controls, we have no indication that individual reference points might systematically differ along the levels of SES, which might introduce bias. According to validation work on survey questions about subjective welfare, reference points between the rich and the poor are not likely to differ [49].

The previous point also touches upon the final limitation regarding the issue of causality. In our analysis, we treat SES as exogenous and assess its influence on health trajectories. Previous research, however, suggests that SES can be affected by changes in health [30]. As the core of this analysis is the change in health through migration and crisis, we are interested in the changes caused by this experience, rather than the question of whether precrisis health caused higher SES in Syrians. A potential issue might arise if temporary periods of good health would induce a rise in SES, which could then level over time following a regression to the mean. However, given the strong and persistent evidence of the relationship between SES and health, we do not think that this dynamic is a main driver of our results.

## Conclusion and future research

The aim of this study was to assess the importance of SES in Syrians before becoming a refugee in Germany. Our results based on a uniquely rich refugee dataset are in line with the concept of status inconsistency, as we only observe a decline in health satisfaction for individuals from a relatively high economic rank. Hence, high SES in the precrisis provides limited protection against the adverse health effects of migration and is associated with a stronger decline in health.

These novel findings bear several practical and research-relevant implications. People from different SES face unique challenges and threats to their well-being and mental health. While people from poorer parts of the Syrian society seem to be less affected by migration and might even benefit from the current conditions as a refugee in Germany, the same conditions appear to be detrimental for individuals previously living in higher classes of the Syrian society. Overall, our data indicate a strong leveling of health indicators and life satisfaction through the cascade of events caused by the conflict in Syria.

Our results are based on data covering mostly a period of 1 to 2 years after migration. Future research should build on the next waves of this dataset to study the influence of premigration SES on health in the longer run, i.e., when most of the refugees enter the labor market or when the cumulative exposures translate into chronic diseases. The ensuing results

might allow for additional insights into the set-point theory [50] and the ability to adapt to novel situations under such extreme conditions [51], which has been shown to be sensitive to the cultural context [52]. Future research could supplement our findings by focusing on the identification of further causal pathways underlying the observed dynamic between SES and health. Such deeper insights might guide policy makers in their search for better regulations for refugees.

## Supporting information

**S1 Table. Correlation between health indicators.**
(DOCX)

**S2 Table. Descriptive statistics by SES.** SES, socioeconomic status.
(DOCX)

**S3 Table. Mental health, SES, and migration experience.** SES, socioeconomic status.
(DOCX)

**S4 Table. Replication of Table 2 using an ordered logit model.**
(DOCX)

**S5 Table. Replication of Table 3 presenting detailed results on covariates.**
(DOCX)

**S6 Table. Replication of Table 3 using an ordered logit model.**
(DOCX)

**S7 Table. Regression underlying Figs 3 and 4.**
(DOCX)

**S8 Table. Life satisfaction similar to Table 3.**
(DOCX)

**S9 Table. Tests for the importance of ceiling effects.**
(DOCX)

**S1 Fig. Relationship between health satisfaction and other indicators.**
(DOCX)

**S2 Fig. Relationship between health worries and health satisfaction.**
(DOCX)

**S1 STROBE Checklist.**
(DOCX)

**S1 Data. Analyses.**
(DO)

**S2 Data. Codebook.**
(DOCX)

## Acknowledgments

The authors would like to thank the following groups and individuals for their valuable comments on this manuscript, in no particular order: all participants of the CBS CBIG brown bag

seminar on 22 May 2019; Alfonso Sousa-Poza, Florence Samkange-Zeeb, and Irwa Issa, particularly for his invaluable comments concerning the situation in Syria.

## Author Contributions

**Conceptualization:** Jan Michael Bauer, Tilman Brand.

**Data curation:** Jan Michael Bauer, Tilman Brand.

**Formal analysis:** Jan Michael Bauer, Tilman Brand.

**Funding acquisition:** Hajo Zeeb.

**Methodology:** Jan Michael Bauer, Tilman Brand.

**Resources:** Hajo Zeeb.

**Supervision:** Hajo Zeeb.

**Validation:** Tilman Brand.

**Writing – original draft:** Jan Michael Bauer, Tilman Brand.

**Writing – review & editing:** Hajo Zeeb.

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
