## [Decision Letter · Decision Letter 0]

15 Jan 2020

Dear Dr. Zeeb,

Thank you very much for submitting your manuscript "Pre-migration socioeconomic status and post-migration health: evidence from Syrian refugees in Germany" (PMEDICINE-D-19-03328) for consideration at PLOS Medicine. 

[LINK]

In light of these reviews, I am afraid that we will not be able to accept the manuscript for publication in the journal in its current form, but we would like to consider a revised version that addresses the reviewers' and editors' comments. Obviously we cannot make any decision about publication until we have seen the revised manuscript and your response, and we plan to seek re-review by one or more of the reviewers. 

We expect to receive your revised manuscript by Jan 29 2020 11:59PM. Please email us (plosmedicine@plos.org) if you have any questions or concerns.

We look forward to receiving your revised manuscript. 

Sincerely,

Clare Stone, PhD

Managing Editor 

PLOS Medicine

plosmedicine.org

Please revise your title according to PLOS Medicine's style. Your title must be nondeclarative and not a question. It should begin with main concept if possible. "Effect of" should be used only if causality can be inferred, i.e., for an RCT. Please place the study design ("A randomized controlled trial," "A retrospective study," "A modelling study," etc.) in the subtitle (ie, after a colon).

Please structure your abstract using the PLOS Medicine headings (Background, Methods and Findings, Conclusions).

Abstract Background: Provide the context of why the study is important. The final sentence should clearly state the study question.

Abstract Methods and Findings:

* Please ensure that all numbers presented in the abstract are present and identical to numbers presented in the main manuscript text.

* Please include the study design, population and setting, number of participants, years during which the study took place, length of follow up, and main outcome measures.

* Please quantify the main results (with 95% CIs and p values).

* Please include the important dependent variables that are adjusted for in the analyses.

* Please include the actual amounts and/or absolute risk(s) of relevant outcomes (including NNT or NNH where appropriate), not just relative risks or correlation coefficients. (example for absolute risks: PMID: 28399126). 

* Please include a summary of adverse events if these were assessed in the study.

Data – the URL you provide does not seem to take one to a repository. All data needs to be made available – and for referees to also assess. Please ensure this is available on resubmission. Noting an author cannot be a point of contact. Also remove the reference to data and on request from line 102.

Please ensure all questionnaires are provided as Supp files (and translated into English, as needed). 

Line 249 – where (not were)

All of your Figs appear to have converted oddly and appear very small. Please do visit our guidleines on figures format and size. 

Please ensure that the study is reported according to the [STROBE] guideline, and include the completed [STROBE or other] checklist as Supporting Information. When completing the checklist, please use section and paragraph numbers, rather than page numbers. Please add the following statement, or similar, to the Methods: "This study is reported as per the Strengthening the Reporting of Observational Studies in Epidemiology (STROBE) guideline (S1 Checklist)."

Please report your study according to the relevant guideline, which can be found here: http://www.equator-network.org/

Comments from the reviewers:

Reviewer #1: "Pre-migration socioeconomic status and post-migration health: evidence from Syrian refugees in Germany" analyzes survey data of 2000-plus Syrian refugees that had entered Germany on or after 2013, using two main standard multivariate regression models - the first a cross-sectional analysis for five health measures for their current status in Germany (T1), and the second a longitudinal analysis also involving their condition when in Syria (T0).

The authors modify both main regression models with varying combinations of covariates (results summarized in Tables 2 & 3) such as sex, age, sociodemographics, migration experience & experience in Germany, which generally produce similar (robust) outcomes, with the major exception of sex (covered in the section on Heterogeneity). The key conclusion is that the main dependant variable of after-migration "health satisfaction" is strongly attentuated (significantly less strongly correlated) with reference to pre-migration SES, in contrast to pre-migration "health satisfaction" (expectedly?) being highly correlated with SES, as visually displayed in Figures 1 & 2.

1) A first observation would be that the conclusions from the current analysis might more properly be claimed on "post-migration health satisfaction" rather than "post-migration health" (as in the manuscript title), as the relevant longitudinal analysis was performed on "health satisfaction" and not "(self-reported) health", notwithstanding the relatively high (~0.8) correlation between the two indicators as shown in Table A1. Moreover, the correlation with "mental distress (mental health?)" is weak.

The distinction between "health satisfaction" and (actual) "self-reported health" would appear to be important, because there likely remain interactions between expressed satisfaction and underlying health that are difficult to quantify (e.g. one's health is so-so, but he thinks it reasonable given his current state as a refugee, and thus expresses satisfaction). There might be latitude to visualize the typical relationship between "health satisfaction" and "self-reported health" in a manner similar to Figures 1 & 2.

2) Following on, a possible approach for analyzing "self-reported health" longitudinally, might be to substitute the missing T0 "self-reported health" data, with values predicted using T0 "health satisfaction" and the relationship between T1 "health satisfaction" and T1 "self-reported health", assuming this relationship stays constant.

3) Is there a particular justification for adding the "sociodemographics", then "migration experience", then "experience in Germany" covariates in that order, for both the cross-sectional and longitudinal models?

4) The authors note possible ceiling effects of a finite scale in their discussion of limitations. It is known that individuals may subjectively treat the scales in different ways (e.g. some might use only the top few values, while others might use the full scale). Might any of the (possibly unused) survey responses be used to estimate such tendencies? The authors may wish to discuss this possible effect further.

5) In Equation 1, what does the Beta_1a term refer to?

6) There are some minor grammatical/phrasing issues, e.g. "...the fact that migrant often live" (Line 14), "During migration passage... but also their legal status, housing conditions, economic prospects and the contexts of reception." (Line 23-28); also there is an occasional inconsistency with citation practices, for example "Spallek and colleagues (2011)..." in Line 20, with appears without the standard bracketed reference to [8] and/or [9].

Reviewer #2: Review PLoS Medicine: Pre-migration socioeconomic status and post-migration health: evidence from Syrian refugees in Germany.

This is a paper that is of high interest for people in the field and policy-makers. It is well-written and well-conducted. I have some comments that could possibly ameliorate this paper.

Major Comments:

- Please add some references to the first paragraph.

- p. 2, line 51: please explain why did you then decide to use individual SES and not at the household level?

- Please explain the S&W concerning the use of self-reported SES, e.g. over-or underestimation,…

- p. 6: you include education as a covariate but isn't it a measure of SES?

- please explain why you didn't stratify all the analyses by gender as both self-reported SES and self-reported health may differ by gender.

- Please discuss the age limits of the study.

- p. 9: please explain the category of education 'certificate different'.

- p. 9: please discuss gender differences in the descriptives as well.

- The tables should be readable and easily interpretable on itself. Please explain in the titles and headings what numbers are shown.

- The figures are not readable.

- As the dependent variables are self-reported health variables, I personally think that these observed differences are not so much related to access to health care but more to experiences along the way. Please discuss this issue in the discussion paragraph.

- p. 17, line 352: as this is a cross-sectional study, please be careful when using words as 'influences'. 

Minor Comments:

- p. 2, lines 28-30: please check the construction of this sentence.

- p. 2 line 34: please first write SES in full spell before using the abbreviation.

- p. 2, line 45: lead to.

- p. 3, line 58: pathways,

- p. 3, line 76: SES instead of SESs.

- p. 4, lines 96 and 100: data were instead of data was.

- p. 5, line 129: please check the construction of this sentence.

- p. 9, line 201: happened.

- p. 17, line 341: our data suggest.

- p. 19, line 390: specific points in time.

Reviewer #3: Overall this is a useful study using a large dataset, and provides novel findings on the complex relationship between SES and mental health. I have listed minor points below, the majority of which relate to a lack of polished language/descriptions and multiple typos. Once these are addressed, I think this study would be well worthy of publication. 

Abstract

I would suggest avoiding using the word 'influx' as this has become a politicized word

Change 'on the top' to 'at the top'

Please name the study design

Line 8 amend 'their' to 'migrant'

I think it is misleading to refer to a longitudinal analysis when data was only collected at one time point? Could it be referred to as a retrospective analysis?

Introduction 

P1 line 4 - can the authors define what 'top country' refers to? Country with the most migrants?

P1 line 14 - change migrant to migrants

P2 line 34 -please define the acronym in the first use

P2 line 45 - leads instead of lead

P3 line 57 - 'the' chronically ill

P4 line 80 - please state the study design 

P4 line 81 - the question 'of'

P5 line 87 - 'a' scientific 

P4 line 88-91 - the hypotheses could be clearer. What is meant by safeguard? As we don't know the study design or what kind of data will be used, it's hard to know what this means. The first hypothesis also seems to include two hypothesis within it.

Methods

P4 line 93 - again, the study design needs to be explicit from the start, before describing where the data comes from

P4 line 93 - international readers, like me, will not be familiar with the SOEP. Could the authors add a sentence to describe what it is? 

P4 line 95 - why predominantly in these dates? Can the authors be specific about which dates are included?

P4 line 99 - change 'their' to migrant'

P4 line 103 the link is helpful but I think more detail should be in this section. How were the migrants identified by the system? How were they first contacted? How was an interpreter arranged? As it stands, it is quite unclear how these data was generated. It is also unclear what the overall pupose of data collection was, and exactly what was included. 

P5 line 114 - as 'an' indicator

P5 line 120 - 'used' not 'use'

P5 line 124 - 'was' measured

P5 line 126 - it might be worth citing studies that show the PHQs validity in migrant populations too? https://www.ncbi.nlm.nih.gov/pubmed/30156742

P6 line 140 - sentence doesn't make sense

P6 line 155 - for a 'full' description

P7 line 157 - change 'their' to 'migrant'

Results

P9 table 1 - the table is a little unclear, what does Income N/A mean and why is the % so high? Given that income is a core part of SES it seems an important variable to be missing almost half of?

P10 line 210 - sentence does not make sense

P12 line 244 - rephrase 'lower levels of health' Its unclear what this means

P12 line 248 - rephrase pre-migration 'area'

Discussion

P17 line 335 - people who have been trafficked instead of traffic persons

P17 line 342-43 - rephrase as it does not make sense

P18 line 381 - the two time points seems misleading, isn't it retrospective? The wording here (and in thr abstract) implies to points of administration of measures? It's also unclear how missing data was handled. 

P19 line 389 - 'a' point in time

P19 line 404 - 'Syrians' or 'Syrian migrants' not 'Syrian'

[LINK]

---

## [Decision Letter · Decision Letter 1]

18 Feb 2020

Dear Dr. Zeeb,

Thank you very much for re-submitting your manuscript "Pre-migration socioeconomic status and post-migration health satisfaction among Syrian refugees in Germany: a retrospective analysis" (PMEDICINE-D-19-03328R1) for review by PLOS Medicine.

I have discussed the paper with my colleagues and the academic editor and it was also seen again by one of the original reviewers. I am pleased to say that provided the remaining editorial and production issues are dealt with we are planning to accept the paper for publication in the journal.

[LINK]

We look forward to receiving the revised manuscript by Feb 25 2020 11:59PM. 

Sincerely,

Clare Stone, PhD

Managing Editor 

PLOS Medicine

plosmedicine.org

Requests from Editors:

Data- the link www.diw.de appears to be a general site. It is not clear how one can access data from this. Please provide a direct link to the data from your study. Also data on request (one presumes from the authors) does not comply with our open data policy where authors cannot be points of contact. Please provide the contact details of say a data committee where the method/ code can be accessed from. 

Please substitute "cross-sectional" for "retrospective" in the title

- demographic details are needed in the abstract as well as mean age. 

- p<0.01 -> exact value or p< 0.001 please

Please change gender to sex. The terms gender and sex are not interchangeable (as discussed in http://www.who.int/gender/whatisgender/en/ ); please use the appropriate term

- introduction is very long – please shorten to improve accessibility

Did your study have a prospective protocol or analysis plan? Please state this (either way) early in the Methods section.

c) In either case, changes in the analysis—including those made in response to peer review comments—should be identified as such in the Methods section of the paper, with rationale.

I cannot see a table with absolute numbers of participants and their demographic information. Please provide this in the main text and ideally this should be table 1 and re-order others tables accordingly. 

STROBE – please be accurate and provide sections instead of ‘first part’. 

Comments from Reviewers:

Reviewer #1: The authors have addressed our original points satisfactorily; the additional clarifying analyses/charts were appreciated.

Given the emphasis on the 5 SES categories in the analysis (as also illustrated in Figures 1 to 4), it might be appropriate to include a breakdown of characteristics (as from Table 1) by these SES categories, including the number of subjects in each category.

The conflation of "health" and "health satisfaction" throughout the text remains slightly concerning; authors might consider re-emphasizing the use of health satisfaction from surveys as a proxy for health, at opportune points in the manuscript (e.g. the beginning of the discussion section)

[LINK]

---

## [Editor Report · Decision Letter 2]

28 Feb 2020

Dear Prof. Zeeb, 

On behalf of my colleagues and the academic editor, Dr. Paul Spiegel, I am delighted to inform you that your manuscript entitled "Pre-migration socioeconomic status and post-migration health satisfaction among Syrian refugees in Germany: a cross-sectional analysis" (PMEDICINE-D-19-03328R2) has been accepted for publication in PLOS Medicine. 

PRODUCTION PROCESS

PRESS

PROFILE INFORMATION

Thank you again for submitting the manuscript to PLOS Medicine. We look forward to publishing it. 

Best wishes, 

Clare Stone, PhD

Managing Editor 

PLOS Medicine

plosmedicine.org